# Optimization of Mixed Energy Supply of IoT Network Based on Matching Game and Convex Optimization

**DOI:** 10.3390/s20195458

**Published:** 2020-09-23

**Authors:** Dongsheng Han, Tao Liu, Yincheng Qi

**Affiliations:** Department of Electronic and Communication Engineering, North China Electric Power University, Baoding 071003, China; taoliu1995@163.com (T.L.); qiych@ncepu.edu.cn (Y.Q.)

**Keywords:** IoT, 5G, matching game, convex optimization, smart homes

## Abstract

The interaction capability provided by the Internet of Things (IoT) significantly increases communication between human and machine, changing our lives gradually. However, the abundant constructions of 5G small base stations (SBSs) and large-scaled access of IoT terminal equipment (TE) will surely cause a dramatic increase in energy expense costs of a wireless communication system. In this study, we designed a bilateral random model of TE allocation and energy decisions in IoT, and proposed a mixed energy supply algorithm based on a matching game and convex optimization to minimize the energy expense cost of the wireless communication system in IoT. This study divided the problem of minimizing energy expense cost of the system into two steps. First, the random allocation problem of TEs in IoT was modeled to a matching game problem. This step is to obtain the TE matching scheme that minimizes the energy consumption of the whole system on the basis of guaranteeing the quality of service of TEs. Second, the energy decision problem of SBS was modeled into a convex optimization problem. The energy purchase scheme of SBSs with the minimum energy expense cost of the system was obtained by solving the optimal solution of the convex optimization. According to the simulation results, the proposed mixed energy supply scheme can decrease the energy expense cost of the system effectively.

## 1. Introduction

The concept of the Internet of Things (IoT) was proposed officially at the end of the 20th century. In recent years, the development of various communication technologies has provided technical support for the wide application of the IoT. We can divide these technologies into three categories based on transmission rates and coverage. ZigBee and Wi-Fi technologies applied in local area networks are suitable for providing short distance communication services; 5G cellular network technology is suitable for high-rate transmission scenarios. Low-power wide-area networks technology represented by LoRa can provide low power and long-distance communication with low transmission rate [1]. With the gradual construction of the space-terrestrial integrated networks [2], the IoT will bring extensive interaction between man and machine and between machine and machine. Thus, the IoT possesses promising applications in various fields, including the smart home, automatic driving, and medical field, and will become an indivisible part of our daily lives in the near future [3]. As more and more smart devices are introduced into family life, the smart home provides efficient management solutions for families in many fields, such as security, fire protection, and lighting [4].

However, communication flows of cellular network will be doubled generally every year [5]. Data statistics show that the power consumption of base stations (BSs) accounts for approximately 50%–80% of the total power consumption for the information and communication technological industry [6,7]. Massive IoT nodes and connections require lower latency, higher reliability, and other transmission features than ever before [8], which will surely cause a dramatic increase in the energy expense cost of the wireless communication system. For communication operators, the energy expense cost of BSs is roughly 18%–32% of the total operation cost [9,10]. The energy purchased by communication operators is mainly electric energy from thermal power generation. With the intensifying trend of global warming, people increasingly reach an agreement on power supply for the IoT based on renewable energy sources. Moreover, renewable energy sources are superior to fossil energy in terms of ecology and economy [11].

With the increasing demand of various new services and real-time services, wireless networks need to support requirements with different traffic characteristics and different quality of service (QoS) guarantees [12]. In order to ensure that the terminal equipment can always maintain the best connection, it is important to design an efficient handover mechanism. Vertical handover (VHO) can achieve seamless mobility and service continuity between various mobile wireless access networks [13]. Lai WK et al. [14] proposes a new handover management scheme for D2D communication in 5G network, which reduces handover times and handover latency. Seo D.Y. and Chung Y.W. [15] has made efforts to optimize BSs energy consumption and reduce the number of handovers in 5G network, and is committed to achieving green communication in 5G network.

The development of smart grids provides a new solution to the green development of the 5G network. The smart grid constructs an extensively distributed automatic energy transport network through two-way information and energy flows, so that BSs can flexibly apply energy sources of different energy suppliers in the smart grid [16,17]. Xu and Zhang [18] studied the energy supply problem of BS in a coordinated multiple-point smart grid for intelligent power supply, optimized the two-way energy exchange among BSs with the local renewable energy power generation and the smart grid through convex optimization algorithm, and realized the minimum energy expense cost of the system under a given QoS of terminal equipment (TEs). However, the influences of different energy prices of energy suppliers in the smart grid on the energy expense cost of BSs were neglected. Ghazzai and Kadri [19] considered dynamic pricing of electricity of different energy suppliers in the smart grid. In a cellular network, the energy expense cost is minimized by optimizing the energy purchasing schemes from all suppliers. Although this study considered different energy suppliers in the smart grid and varying service types in BSs, the influences of different service qualities of TEs in the same service type on total power consumption of BSs were ignored.

With the continuous development of a smart grid, the renewable energy generation unit of BSs provides an opportunity to decrease the energy expense cost of a communication system [20]. Rached et al. [21] proposed an energy purchasing strategy of the cellular network to solve the uncertainty in renewable energy supply. This strategy involves simultaneous power supplies of the smart grid and uncertain local renewable energy sources. Xin et al. [22] proposed a new online control technique over the communication system in the smart grid based on random subgradient by using the advanced time-decoupling technology to solve the high randomness problem of renewable energy sources. This online control technique realized a feasible and asymptotically optimal resource scheduling when the statistics of random process is unknown. Although these two studies solved the uncertainty problem of renewable energy generation, they still neglected the influences of QoS of accessed TEs on power consumption of BSs.

In addition, applications of coordinated multipoint transmission technology provide a new solution to the energy optimization of communication systems. Xu et al. [23] proposed a suboptimum zero-forcing precoder to solve the energy-saving precoding problem in heterogeneous network and realized energy efficiency optimization that meets QoS constraint of TEs and maximum transmission power constraint of BSs under multiple interferences. However, this suboptimum zero-forcing precoder only realized the optimization of relative energy efficiency and did not consider the absolute energy expense cost of BSs. Wang and Zhu [24] proposed a power control algorithm of micro BSs based on the game theory, which realized maximum total rate of cellular network, to solve the power control problem of BSs in a heterogeneous cellular network under multiple interfaces. Although this study realized the maximum total rate of a cellular network under power constraint, the influences of energy suppliers with different prices on the operation network of a cellular network were neglected. Xu and Zhang [25] designed a coordinated multipoint wireless communication system that has a power supply through renewable energy generation, aiming to maximize the communication and rate of all TEs. Given the new power constraint of transmission loss, the sum of the communication rate of all TEs was realized by jointly optimizing the transmitted power allocation and energy exchange of cooperated BSs. Although a new joint communication and energy cooperation scheme was proposed, BSs in this study were powered by using renewable energy sources, where energy sources that were brought by BSs from the power grid were neglected.

Existing studies on the energy consumption problems of BSs mainly focus on the following two aspects. On the one hand, the QoS of TEs is optimized in these studies under the total energy limits of BSs assuming that the energy source of BSs is provided by one supplier. On the other hand, given a fixed QoS of TEs, the energy supply ratio of different suppliers is optimized. However, no comprehensive considerations have been given to situations on the energy and communication sides. Different QoS demands of TEs in BS and energy sources that BSs purchased from energy suppliers are a bilateral random problem. Changes in sending power caused by different QoS demands of TEs can influence the quantity of energy sources that BSs purchase from energy suppliers, and vice versa.

This study targets at minimum energy expense cost of the wireless heterogeneous communication system in IoT. Hence, a bilateral random system model was designed. Based on this system model, a mixed energy supply algorithm of matching game and convex optimization was proposed, which can realize the minimum energy expense cost of the system. The main contributions of this study are introduced as follows:

1. This study designed a bilateral random system model of TE allocation and energy decisions in IoT to minimize the energy expense cost of the wireless heterogeneous communication system in the IoT. In the system model, TEs with different QoS demands are distributed within the service scope of small base stations (SBSs) randomly, and SBSs purchase energy sources from different energy suppliers who have varying selling prices. Based on this model, a mixed energy supply method based on matching game and convex optimization was proposed. During the allocation of TEs in IoT, a new matching game model was designed to depict the mutual selection behaviors of SBSs and TEs. For the energy purchase decision problem of SBS in IoT, the energy purchase scheme with minimum energy expense cost of the system was gained based on the optimal energy selection mechanism of convex optimization algorithm.

2. When allocating TEs in IoT, this study hypothesized that TEs in the service scope of SBSs have different QoS demands. The allocation problem of TEs was modeled into the Mixed-Integer Nonlinear Programming (MINP). The optimal allocation scheme of TEs was solved by using the matching game model. In the matching game model, the utility functions of SBSs and TEs were built to describe the preferences of SBSs (TEs) to TEs (SBSs). Based on the guaranteed QoS of TEs, the TE allocation scheme with the minimum energy consumption in the whole system was gained through the game between SBSs and TEs.

3. When making energy decisions for SBSs in IoT, the energy selection problem of SBSs in IoT was modeled into a convex optimization problem. Different energy prices and maximum energy supply capacities of energy suppliers were introduced into the convex optimization problem, thereby obtaining the energy decision for the minimum energy expense cost of the system. According to the simulation results, the proposed algorithm effectively decreases the energy expense cost of the system by optimizing the TE allocation and energy purchase schemes of SBSs, while meeting the QoS demands of different TEs and energy constraints of SBSs.

The remainder of this study is organized as follows. Section 2 introduces the system model and problem modeling. Section 3 introduces the matching game model and matching exchange algorithm of TEs. Section 4 introduces the energy decision problem of the system. Section 5 describes a simulation analysis of the proposed algorithm and compares it with the traditional algorithm. Section 6 concludes the study.

## 2. System Model and Problem Modeling

### 2.1. System Model

A wireless heterogeneous communication network system with N SBSs, which serves for K TEs in the IoT, is shown in Figure 1. Each TE is equipped with one antenna, and each SBs has large-scaled multiple-input-multiple-output (MIMO) antenna in the system. In our study, the macro base station is only powered by traditional energy, and each SBS is powered by a hybrid of its own renewable energy devices and energy suppliers in the smart grid. Since we only focus on the TE allocation and energy optimization of the SBS, the impact of the macro base station on the system is ignored in this paper. In this system, the set N={1,2,…,N} is used to represent all SBSs, and the set K={1,2,…,K} is applied to express all TEs. The $BS_{n}$ serves for $K_{n}$ TEs, and $\sum_{i=1}^{N}K_{i}=K$. In the system, TEs distribute randomly within its service scope and can be provided by different SBS. Each TE can obtain services from only one SBS to assure QoS of TEs and avoid wasting communication resources. In this study, $R_{k},k\in K$ is applied to express a bit rate needed by the TE k. Given that different TEs require various service types, each TE has the corresponding QoS demands. For different TEs, $R_{k}\neq R_{k^{\prime }},k,k^{\prime }\in K,k\neq k^{\prime }$. In a practical system, many modes exist for SBSs to collect charges from terminals. In this study, SBSs may gain profits and maintain their energy expense costs by collecting service fees from TEs. Different SBSs have varying prices to unit code rate, and the unit code rate price that SBS n collect from the TE k is $\gamma_{n},n\in N$. Each SBS is equipped with a renewable energy generation unit and purchases energy from energy suppliers in a smart grid to offset energy shortage. In a smart grid, L energy suppliers have different prices, and the energy supplier l provides energy sources to SBSs through the smart grid. In a system, the intelligent control center can gain bilateral information flows between it and SBSs and between it and energy suppliers and makes decision on TE allocation and energy purchase by collecting information. For the convenience of calculation, the energy mentioned in this study refers to the energy in unit time.

We assume that TEs distribute randomly in the service scope of the whole IoT. Each TE k that belongs to the set K is allocated to an SBS n,∀n∈N and served by this SBS. The matrix $V_{N\times K}=\left[\nu_{n,k}\right],\forall k\in K,\forall n\in N$ is applied to express the allocation situation of all TEs, where $\nu_{n,k}$ is defined as the situation that the SBS n serves the TE k. $\nu_{n,k}$ is expressed as:(1)$$\nu_{n,k}=\left\{ \begin{array}{l} 1,SBS\;n\;\mathrm{serves}\;\mathrm{the}\;\mathrm{TE}\;k\\ 0,else \end{array}\right..$$

The set $K_{n}=\left\{k\in K|\nu_{n,k}=1\right\},\forall n\in N$ is the set of TEs that are allocated to the SBS n. Considering the uniqueness of SBS serving TE, $K_{n}\cap K_{n^{\prime }}=Ø,n,n^{\prime }\in N,\mathrm{and}\;n\neq n^{\prime }$. $\nu_{n,k}$ must meet the following equation to assure that each TE k is served by an SBS:(2)$$\sum_{i=1}^{N}\nu_{i,k}=1,\forall k\in K.$$

The SBS adjusts its transmitted power to TEs to assure their service qualities. The power consumption of SBS n can be divided into two parts. One is the signal transmitted power that is determined by using the quantity of accessed TEs:(3)$$P_{e}=\sum_{i=1}^{K}\nu_{n,i}p_{n,i},\forall n\in N,$$
where $p_{n,i}$ is the transmitted power of an SBS to the TE i, and its value is related to the QoS of users, which will be introduced in detail in the following text. The other part is the fixed power $P_{fix}$ that is unrelated with the quantity of accessed TEs. This variable is the power needed to meet the encoding/decoding hardware of SBSs, cooling equipment, and other facilities.

Here, channel gain of SBS n to TE k is expressed by $\eta_{n,k}$. Given that only the influences of large-scaled attenuation on the information channel received attention, it can be gained by:(4)$$\eta_{n,k}={\left(\frac{\lambda }{4\pi }\right)}^{2}\frac{1}{{\left(d_{k}^{n}\right)}^{2}}G,\forall k\in K,\forall n\in N,$$
where $d_{k}^{n}$ is the distance between SBS n and TE k, λ is the wavelength of the sending signals, and G is gain of sending and receiving antennas. Here, $\beta ={\left(\frac{\lambda }{4\pi }\right)}^{2}G$. The TEs are also disturbed by white noises with a power spectral density of $n_{0}$ in the information channel. In the system, the channel bandwidth that is allocated to an SBS is B. Therefore, the bit rate of TE k in the SBS n can be expressed as:(5)$$R_{k}=\log_{2}\left(1+\frac{\eta_{n,k}p_{k}}{n_{0}B+I}\right),$$
where I is the interuser interferences to the TE k. Given that each SBS in the system is equipped with massive MIMO antennas and interuser interferences are eliminated by using zero-forcing precoding technology [26], Equation (5) can be rewritten as:(6)$$R_{k}=\log_{2}\left(1+\frac{\eta_{n,k}p_{k}}{n_{0}B}\right).$$

In this study, the set of energy suppliers is expressed by L={1,2,…,L}. Each SBS is equipped with a renewable energy generation unit, and the renewable energy generation power of the SBS is expressed by $g_{RE}$. $g_{RE}$ can be obtained using the Markov model mentioned in [27]. Additionally, SBSs purchase energy from energy suppliers in the smart grid to offset shortages of renewable energy supply. The energy quantity $x_{n}$ that the SBS n purchases from the power grid in unit time is defined as:(7)$$x_{n}=\sum_{i=1}^{K}\nu_{n,i}p_{n,i}+P_{fix}-g_{RE},\forall n\in N.$$

When an SBS has to purchase energy from different energy suppliers in the smart grid, the energy quantity that the base n purchases from the energy supplier l is $x_{n}^{l},\forall n\in N,\forall l\in L$.
(8)$$x_{n}=\sum_{j=1}^{L}x_{n}^{j}=\sum_{i=1}^{K}\nu_{n,i}p_{n,i}+P_{fix}-g_{RE},\forall n\in N.$$

In the present study, different energy suppliers are hypothesized to have varying selling prices [19]. The energy price that the energy supplier l offered to the SBS is defined $c^{l},\forall l\in L$. The expenses of SBS n to purchase energy can be expressed as:(9)$$f_{n}=x_{n}c^{T},$$
where $c=\left[c^{1},c^{2},\ldots ,c^{l}\right]$ is the energy price vector of energy suppliers, $x_{n}=\left[x_{n}^{1},x_{n}^{2},\ldots ,x_{n}^{l}\right],\forall n\in N$ is the energy purchase scheme of SBS n, and $f_{n}$ is the expense of the SBS n to purchase energy. Therefore, the matrix $X_{L\times N}=\left[x_{1}{}^{T},x_{2}{}^{T},\ldots ,x_{n}{}^{T}\ldots ,x_{N}{}^{T}\right],\forall n\in N$ can be used to express energy purchase information of all SBSs from different energy suppliers in the system.

### 2.2. Problem Modeling

The energy expense cost of the whole system is minimized by optimizing the TE allocation matrix V and energy purchase matrix X in an SBS. The energy expense cost of the system can be expressed as:(10)$$f=\sum_{n=1}^{N}x_{n}c^{T},\forall n\in N,$$

In the system, the transmitted power $p_{n,k}$ of the SBS n provided to the TE k can be rewritten from Equation (6) to assure the QoS of TEs:(11)$$p_{n,k}=\nu_{n,k}\frac{1}{\eta_{n,k}}\left(2^{R_{q}}-1\right)n_{0}B,\forall k\in K,\forall n\in N.$$

The SBS has the highest transmitted power limit $P_{e,\max}$. Therefore, the total transmitted power of an SBS must meet:(12)$$\sum_{i=1}^{K}\nu_{n,i}p_{n,i}\leq P_{e,\max},\forall n\in N.$$

Energy suppliers have a maximum energy supply amount, which is expressed by $x_{l}^{\max}$. Therefore, the energy amount that the system purchased from the energy supplier l has the following constraints:(13)$$\sum_{i=1}^{N}x_{i}^{l}\leq x_{l}^{\max},\forall l\in L.$$

In summary, the objective function of minimum energy expense cost of the system can be expressed by combining Equations (1), (2), (4), (8) and (11)–(13) as follows:(14)$$\underset{V,X}{\min}f\left(V,X\right)=\underset{V,X}{\min}\sum_{n=1}^{N}x_{n}c^{T},$$
subject to:(15)$$\eta_{n,k}=\beta \frac{1}{{\left(d_{k}^{n}\right)}^{2}},\forall k\in K,\forall n\in N,$$
(16)$$p_{n,k}=\nu_{n,k}\frac{1}{\eta_{n,k}}\left(2^{R_{q}}-1\right)n_{0}B,\forall k\in K,\forall n\in N,$$
(17)$$\sum_{i=1}^{K}\nu_{n,i}p_{n,i}\leq P_{e,\max},\forall n\in N,$$
(18)$$\sum_{i=1}^{K}\nu_{i,k}=1,\forall k\in K,$$
(19)$$\nu_{n,k}\in \left\{0,1\right\},\forall k\in K,\forall n\in N,$$
(20)$$\sum_{i=1}^{N}x_{i}^{l}\leq x_{l}^{\max},\forall l\in L,$$
(21)$$\sum_{j=1}^{L}x_{n}^{j}=\sum_{i=1}^{K}\nu_{n,i}p_{n,i}+P_{fix}-g_{RE},\forall n\in N,$$

Equation (15) expresses the channel gain of TE, and Equation (16) assures the limiting condition of QoS for TEs. Equation (17) indicates that each SBS has an upper bound of output power, and Equation (18) assures that each TE is served by an SBS. Equation (19) is the value range of binary parameters for “switch” of the TEs, and Equation (20) indicates that each energy supplier has an upper limit of energy generation. Equation (21) demonstrates the relationship between energy quantity that SBS must purchase from the smart grid and energy consumption of the SBS.

Through Equation (14), the system resource can be optimized, and the energy expense cost of the system is minimized. Nevertheless, the following are the difficulties in the optimization process:
1.When the energy quantity purchased by all SBSs is fixed, how to allocate TEs with different QoS demands in the IoT to different SBSs is a Mixed-Integer Nonlinear Programming (MINP). This problem is a nondeterministic polynomial time hard (NP-hard) optimization problem [28]. For a fixed TE allocation scheme $V_{N\times K}^{\left(1\right)}$, calculating the total cost is easy. However, the time complexity is difficult to estimate even for a small IoT system when we want to solve the scheme with the minimum energy expense cost of the system.2.When the TE allocation scheme in SBSs is fixed, how SBSs in the IoT purchase energy sources from different energy suppliers is also an NP-hard optimization problem. Similarly, the time complexity is difficult to estimate by solving the global optimal solution through conventional methods.

Hence, a mixed energy supply optimization algorithm based on a matching game and convex optimization is proposed. The objective function in Equation (14) is divided into two parts. In this study, the proposed mixed energy supply optimization algorithm is solved by using two steps. First, we hypothesize that the energy strategy of SBS is fixed, and the TE allocation scheme V that achieves the optimal utility function of the whole system is gained through a multiple-to-one matching game model. Second, the energy decision X of SBSs to realize the minimum energy expense cost of system is gained through a convex optimization algorithm.

When solving the TE allocation problem in the IoT, we hypothesize that the energy decision X of SBSs is known and fixed, and the TE allocation scheme with the minimum energy consumption of the system is acquired by changing service SBSs to TEs. Under the premise of QoS of TEs and the power constraint of SBSs, a utility function of the system is constructed based on the matching game model and expressed as Equation (22):(22)$$\underset{V}{\max}\phi \left(V\right)=\underset{V}{\max}\left(\sum_{k\in K}\phi_{k}+\sum_{n\in N}\phi_{n}\right),$$
subject to:(23)$$\eta_{n,k}=\beta \frac{1}{{\left(d_{k}^{n}\right)}^{2}},\forall k\in K,\forall n\in N,$$
(24)$$p_{n,k}=\nu_{n,k}\frac{1}{\eta_{n,k}}\left(2^{R_{q}}-1\right)n_{0}B,\forall k\in K,\forall n\in N,$$
(25)$$\sum_{i=1}^{K}\nu_{n,i}p_{n,i}\leq P_{e,\max},\forall n\in N,$$
where $\phi_{k}$ and $\phi_{n}$ are the utility function of TE k and the SBS n, respectively. The matching scheme that realizes the optimal utility function of a system is obtained through the matching game. The energy decision of SBSs with the minimum energy expense cost of the whole system can be made through convex optimization algorithm. Hence, the objective function can be expressed as:(26)$$\underset{X}{\min}f\left(X\right)=\underset{X}{\min}\sum_{n=1}^{N}x_{n}c^{T},$$
subject to:(27)$$\sum_{i=1}^{N}x_{i}^{l}\leq x_{l}^{\max},\forall l\in L,$$
(28)$$\sum_{j=1}^{L}x_{n}^{j}=\sum_{i=1}^{K}\nu_{n,i}p_{n,i}+P_{fix}-g_{RE},\forall n\in N.$$

## 3. TE Allocation Based on Matching Game

If the energy quantity purchased by different SBSs in unit time is a fixed value, then the transmission power of each SBS is fixed. Under this circumstance, the optimal TE allocation scheme is searched. Therefore, this study utilizes a multiple-to-one matching game model. In this system, TEs with different QoS demands distribute randomly within the service scope of SBSs in the system. Here, the conditions that meet the exchange and termination of the final exchange are defined. In addition, the utility functions of TEs and SBSs are constructed to describe the influences of exchange on SBSs and TEs. An overall utility function of the system is made to compare the advantages and disadvantages of different schemes to describe the total utility of the SBS and TE system in the whole internet. Moreover, an algorithm is designed to obtain the matching scheme with the maximum utility function of the whole system.

### 3.1. Matching Game Model

The matching game can construct models of bilateral allocation problems between two groups of participants according to their preference relationships [29]. In the proposed model, TEs have different preferences to SBSs, such as channel gain from SBSs and fees that SBS collected from served TEs. SBSs have different preferences to TEs, such as gains from service to different TEs and sending power to TEs. Hence, the TE allocation problem in the IoT was naturally modeled into a multiple-to-one game model in the present study.

A matching scheme of TEs was defined as a matching matrix μ that describes the allocation situation of all TEs in the system. Specifically, an SBS serves several TEs, and one TE is only served by one SBS. This scheme can be described as follows:

**Definition** **1.***The matching matrix is*μ*and*μ⊆K⊗N.
(29)$${\left\|\mu \left(k\right)\right\|}_{1}=1,$$(30)$${\left\|\mu \left(n\right)\right\|}_{1}=K_{n},$$*where the vector*μ(k)={n∈N|(k,n)∈μ}*and the vector*μ(n)={k∈K|(k,n)∈μ}.

In Definition 1, Equation (29) indicates that the TE k is served by only one SBS, and Equation (30) indicates that an SBS serves for a total of $K_{n}$ TEs. μ(k)=n indicates that SBS n serves for TE k, that is, $\nu_{n,k}=1$. $\mu (n)=K_{n}$ reflects the set of all TEs that SBS n serves for.

Two utility functions were introduced into the describe preferences of the two groups of participants, which are utility functions of TE and SBS, to measure the quality of matching schemes.

The channel gain of SBS n to the served TEs was defined as $\vec{\eta_{n}}=\left(\eta_{n,1},\eta_{n,2},\ldots ,\eta_{n,K_{n}},\eta_{n,K}\right)$, and the transmitted power vector of SBS n was $\vec{P_{n}}=\left(p_{n,1},p_{n,2},\ldots ,p_{n,K_{n}},p_{n,K}\right)$. During the matching between TEs and SBSs, the channel gain and transmitted power of SBS n will change after the serviced TEs are changed. Therefore, when a determined matching scheme μ was gained, the channel gain and transmitted power vectors of SBS n were defined as $\vec{\eta_{n}^{\mu }}=\left(\eta_{n,1}^{\mu },\eta_{n,2}^{\mu },\ldots ,\eta_{n,K_{n}}^{\mu },\eta_{n,K}^{\mu }\right)$ and $\vec{P_{n}^{\mu }}=\left(P_{n,1}^{\mu },P_{n,2}^{\mu },\ldots ,P_{n,K_{n}}^{\mu },P_{n,K}^{\mu }\right)$, respectively.

The utility function of TE k of SBS n was defined as:(31)$$\max\phi_{k}\left(\vec{\eta_{n}^{\mu }}\right)=\eta_{n,k}-R_{k}\gamma_{n},$$
subject to:(32)$$\eta_{n,k}=\beta \frac{1}{{\left(d_{k}^{n}\right)}^{2}},\forall k\in K,\forall n\in N.$$

Equation (31) expresses the preferences of TE i to SBS n, and these preferences can be composed of two parts. The first part is the improvement of channel gain of TE k by choosing the SBS n, which is a positive factor for TE k to choose SBS n. The second part is the service cost that SBS n collects from the TE k, which is a negative factor for TE k to choose SBS n.

The utility function of SBS n that serves for $K_{n}$ TEs was defined as:(33)$$\max\phi_{k}\left(\vec{P_{n}^{\mu }}\right)=\sum_{i=1}^{K}\gamma_{n}R_{i}-\sum_{i=1}^{K}p_{n,i}^{\mu },$$
subject to:(34)$$\sum_{i=1}^{K}p^{\mu }{}_{n,i}\leq P_{e,\max},\forall n\in N,$$
(35)$$p_{n,k}=\nu_{n,k}\frac{1}{\eta_{n,k}}\left(2^{R_{q}}-1\right)n_{0}B,\forall k\in K,\forall n\in N.$$

Equation (33) expresses the preferences of SBS n to TEs. The first part of Equation (33) expresses the service gains of the SBS n from the served TEs, which is beneficial for the SBS n. The second part is the transmitted power of SBS n needed to assure the QoS needs of TEs, which is disadvantageous for the SBS n.

Equation (31) shows that attention shall be paid to the service cost of TEs to acquire gains in addition to channel gains of different SBSs to TEs to maximize the utility function of TEs. Equation (33) shows that the low transmitted power of SBSs is conducive to SBS because SBS must purchase energy from energy suppliers at different prices or supply energy sources through renewable energy generation. Nevertheless, decreasing the served TEs will decrease service gains of the SBS, thereby decreasing its profits. Hence, service gains and transmitted power of SBSs must be considered in the process of matching game. Therefore, the matching method in reference [30] was chosen and optimized, achieving the optimal results of the following matching methods.

**Definition** **2.***For*$\forall k,k^{\prime }\in K,\forall n,n^{\prime }\in N,(n,k),(n^{\prime },k^{\prime })\in \mu$, *the matching exchange*$\mu_{k}^{k^{\prime }}$*was defined as*$\mu_{k}^{k^{\prime }}=\left\{\mu |(n,k),(n^{\prime },k^{\prime })\notin \mu \right\}\cup \left\{\left(n^{\prime },k),(n,k^{\prime }\right)\right\}$.

Here, two SBSs could mutually exchange one TE that they serve for. In this process, other TEs belonging to these two SBSs remained the same. However, the exchange had a prerequisite of meeting conditions for gain exchange.

The exchange gain was given by Definition 3.

**Definition** **3.***In a matching scheme*μ, *the exchange is called gain exchange only when a pair of exchange*$\langle k,k^{\prime }\rangle$*of TEs meets the following conditions:*(36)$$\forall t\in \left\{k,k^{\prime },\mu \left(k\right),\mu \left(k^{\prime }\right)\right\},\phi_{t}\left(\mu_{k}^{k^{\prime }}\right)\geq \phi_{t}\left(\mu \right),$$(37)$$\exists t\in \left\{k,k^{\prime },\mu \left(k\right),\mu \left(k^{\prime }\right)\right\},\phi_{t}\left(\mu_{k}^{k^{\prime }}\right)\geq \phi_{t}\left(\mu \right).$$

Equation (36) demonstrates that after finishing gain exchange, the utility functions of the two involved SBSs and TEs did not decrease. Equation (37) demonstrates that after finishing the gain exchange, at least one of the utility functions of the two involved SBSs and TEs will increase. Gain exchange conditions also avoid the circulation of equivalent matching exchange because Equation (37) requires the increase of at least one utility function in exchange.

Gain exchange also determined the condition to terminate matching exchange.

**Definition** **4.***When no gain exchange exists in a matching scheme*μ, *this matching scheme is called stable matching.*

When the matching exchange in a system does not meet the condition of gain exchange, the utility functions of SBSs or TEs will decrease if exchange of TEs continues. In the game theory, each gamer is self-interested, and no one will decrease matching exchange of its own utility functions. At this instance, the system gains the optimal matching scheme.

### 3.2. Matching Exchange Algorithm

In this section, a matching algorithm of TEs was designed to obtain the stable matching scheme. The progressive increase and convergence of the matching algorithm were proposed. In the matching algorithm, each SBS and TE can make decisions rationally according to their preferences. A global utility function was built up to compare the advantages and disadvantages of different matching schemes to measure the performances of different matching schemes.

The objective function of the matching algorithm is as follows:(38)$$\underset{V}{\max}\phi \left(V\right)=\underset{V}{\max}\left(\sum_{n\in N}\sum_{k\in K_{n}}\phi_{k}\left(\vec{\eta_{n}^{\mu }}\right)+\sum_{n\in N}\phi_{k}\left(\vec{P_{n}^{\mu }}\right)\right),$$
subject to:(39)$$\eta_{n,k}=\beta \frac{1}{{\left(d_{k}^{n}\right)}^{2}},\forall k\in K,\forall n\in N,$$
(40)$$\sum_{i\in K}p^{\mu }{}_{n,i}\leq P_{e,\max},\forall n\in N.$$
(41)$$p_{n,k}=\nu_{n,k}\frac{1}{\eta_{n,k}}\left(2^{R_{q}}-1\right)n_{0}B,\forall k\in K,\forall n\in N.$$

Equation (38) expresses the global utility function of the whole system. The first term refers to the sum of utility functions of all TEs belonging to different SBSs. The second term is the sum of utility functions of all SBSs. When the objective function gains the optimal value, the corresponding matching scheme μ is the stable matching, which is the desired matrix V.

Algorithm 1 shows the details of the matching algorithm.
**Algorithm 1: Matching game of TE Allocation** 1.Initialize parameters. Let the number of iteration be t=0, the maximum number of iterations be $t_{top}$, and the minimum difference of utility function be e. TEs distribute randomly within service scopes of different SBSs. Therefore, the initial allocation $\mu_{0}$ is gained. 2.**do** 3.choose two TEs $i,i^{\prime }\in K$ belonging to different SBSs $n,n^{\prime }\in N$ at the iteration t=t+1. 4.**if** the exchange $\langle i,i^{\prime }\rangle$ is a gain exchange, 5.**if** the exchange meets the limiting conditions (32), (34), and (35) 6.Updating $\mu_{t}\leftarrow \mu_{k}^{k^{\prime }}$; 7.**else** retaining the matching $\mu_{t}=\mu_{t-1}$ 8.**else if** the exchange $\langle i,i^{\prime }\rangle$ is not a gain exchange 9.Retaining the matching $\mu_{t}=\mu_{t-1}$10.**Until**$t>t_{top}$ or $\left|\phi \left(\mu_{t}\right)-\phi \left(\mu_{t-1}\right)\right|<e$

Next, progressive increase and convergence of the matching game algorithm for TE allocation were proved. In other words, the uniqueness of the solution of the proposed algorithm was proved. Given that the TE exchange between SBSs only influenced the involved SBSs and the TEs belonging to the involved SBSs, only the involved SBS and TEs had to be considered to prove the progressive increase and convergence of the matching game algorithm for TE allocation.

Without loss of generality, any gain exchange $\langle k,k^{\prime }\rangle$ was chosen. In the initial matching scheme μ, $\mu \left(k\right)=n,\mu \left(k^{\prime }\right)=n^{\prime }$. After gain exchange, the matching scheme gained is $\mu_{k}^{k^{\prime }}$. Given that the premise for each exchange is that the exchange is a gain exchange, at least one utility function of the involved TEs and SBSs in the exchange will increase after matching game algorithm for TE allocation. The utility functions of other gamers will not decrease. Thus, the utility functions of TEs and SBSs involved in the exchange will not decrease after matching game algorithm for TE allocation.

Moreover, the number of gain exchange is limited because the quantities of TEs and SBSs are limited. Progressive increase of the matching game algorithm for TE allocation has been proven in the above text. Therefore, the stable matching scheme with the optimal utility functions of all SBSs and TEs can be gained eventually. This phenomenon indicates that the matching game algorithm for TE allocation finally converges with matching schemes at a stable matching.

Additionally, although the optimal stable matching scheme could not guarantee maximization of the utility functions of single TE or SBS, it can realize the maximum utility function of the whole network. Although the utility function of a TE (SBS) fails to reach the maximum, the part that is “sacrificed” by the maximum utility function brings additional utility values of other TEs (SBSs), thereby realizing the maximum utility functions of all TEs and SBSs.

## 4. Energy Decision Based on Convex Optimization

At the energy side, this study assumed that TE allocation to each SBS is determined and known (the scheme μ is assumed). The optimal energy purchase scheme of SBSs was searched by fixing the energy quantity that each SBS must purchase. During the modeling of energy decision side, Equation (14) and its constraints were rewritten to change the energy decision modeling problem into a convex optimization problem. The global optimal solution of the convex optimization problem was solved by using the optimization algorithm.

Given that the optimal matching scheme μ that meets the stable matching was known, the total energy quantity $x_{n}$ that has to be purchased for each SBS n was known and $x_{n}$ met Equation (8).

Then, Equation (14) and its constraints can be expressed as:(42)$$\underset{X}{\min}f\left(X\right)=\underset{X}{\min}\sum_{n=1}^{N}x_{n}c^{T},$$
subject to:(43)$$\sum_{i=1}^{N}x_{i}^{l}\leq x_{l}^{\max},\forall l\in L,$$
(44)$$\sum_{l=1}^{L}x_{n}^{l}\leq x_{n},\forall n\in N,$$
where $x_{n}=\sum_{i=1}^{K}\nu_{n,i}p_{n,i}+P_{fix}-g_{RE},\forall n\in N$.

After given $x_{n}$, it can be proved that the problem constituted by Equation (42) and its constraint conditions (43) and Equation (44) is convex optimization problem [31]:
1.Since the objective function Equation (42) has a non-positive second derivative with respect to any variable $x_{n}$, Equation (42) is a convex function;2.Since the constraint conditions (43) and (44) are both affine functions, the constraint conditions are convex functions.

Given that the problem expressed by Equation (42) is equal to the convex optimization problem, an optimal solution $X_{L\times N}$ is necessary to minimize the function. Therefore, the optimal solution can be gained through the existing convex optimization algorithm in MATLAB.

Therefore, the solving algorithm of energy decisions of SBSs can be gained (Algorithm 2).
**Algorithm 2: Algorithm for Minimum Cost of Energy Decision**1.Initialize parameters2.Implement the matching game algorithm for TE allocation and gain the optimal stable matching scheme $V_{N\times K}$.3.Gain the desired purchase power of SBSs from Equation (7)4.The optimal bilateral stable matching scheme $V_{N\times K}$ is brought into Equation (14), and a new model is constructed (Equation (42)).5.Recall the CVX function in MATLAB to gain the minimum optimal energy decision $X_{L\times N}$

## 5. Results

In this section, a simulation study on the proposed bilateral random algorithm was carried out. The mixed energy supply algorithm was compared with the traditional algorithm. The energy expense costs of systems that were calculated through two algorithms under different quantities of TEs and different prices of energy suppliers were judged by using the energy expense cost of a system as the judgment standards. In the traditional algorithm, the system allocated TEs to different SBSs according to the distance between SBSs and TEs. With respect to energy purchase, the system purchased energy from all energy suppliers in average. All simulations are made using MATLAB on the personal computing with Intel^®^ Core i7-4790 CPU @ 3.6 Hz and 8 GB RAM.

Here, we hypothesized that the whole IoT system had *N* SBSs, and the service scope of each SBS was a round area with a radius of 25 m. The coordinates of four SBSs in the Cartesian coordinate system were $SBS_{1}=(16,16)$, $SBS_{2}=(-16,16)$, $SBS_{3}=(-16,-16)$, and $SBS_{4}=(16,-16)$. TEs distributed randomly in the service scope of each SBS. In the system, all SBSs purchased energy sources from L energy suppliers. Like [32], we use probability theory to randomly distribute the TEs within the service scope of the system.

Table 1 shows the parameter setting in this study.

Table 2 shows the number of gain exchanges and running time needed for the system to reach stable matching under different numbers of TEs. Given that TEs distributed randomly in the service scopes of SBSs, 100 tests under the same number of TEs were carried out to avoid errors of experimental results caused by such distribution randomness, and the number of average gain exchanges to reach the stable matching was chosen. With the increase of TEs, the number of gain exchanges that was needed for the system to realize stable exchange achieved a linear growth.

Figure 2 shows the variation curves of utility functions of the system in the matching game algorithm for TE allocation with number of exchanges when the number of TEs is $K_{1}=100$ and $K_{2}=300$. With the increase of number of exchanges, the utility function of a system increased continuously but remained unchanged when the system reached stable matching. In this process, SBSs and TEs accomplished mutual selection through gain exchanges. This finding proves that the proposed matching game algorithm for TE allocation is converging. When the algorithm proposed in [37] serves six terminal devices, the number of iterations required to achieve convergence is five. In contrast, the number of iterations required to achieve convergence of the proposed system is acceptable.

In addition, this study observed how the system applies the mixed energy supply algorithm to reduce energy expense cost when the number of TEs is K=200.

Figure 3 shows the final distribution of TEs after the system implemented the minimum cost algorithm of energy decision under K=200. The connecting lines implied that TEs were served by the connected BSs. The BS that served for TE 1 was changed from the initial $SBS_{4}$ into $SBS_{1}$. This change is because the distances of TE 1 to $SBS_{4}$ and $SBS_{1}$ were similar, and TE 1 preferred $SBS_{1}$ with a lower charge of unit bit rate. In the matching exchange between TE 2 and 3, the latter chose $SBS_{2}$, although it was closer to $SBS_{4}$ because the reduction of service charges to the TE 3 after choosing $SBS_{2}$ was higher than the reduction of the channel gain. For $SBS_{4}$, although the TE 2 increased the transmitted power, the service gains of $SBS_{4}$ were higher than the increase of transmitted power, thereby resulting in successful exchanges. This change will increase the utility function of $SBS_{4}$. After the matching game algorithm for TE allocation was implemented, the total energy consumption of the system decreased by approximately 44% from 554.35 W at the initial state to 308.51 W. The resulting energy consumption was roughly 23% lower than that (403.85 W) of the traditional algorithm of distance-based TE allocation.

Figure 4 depicts the energy purchase schemes of different BSs after the system implemented the minimum cost algorithm based on energy decision under K=200. Table 3 shows the energy prices and maximum energy supply amounts of different energy suppliers. According to the simulation results, BSs in the system preferred energy suppliers with low prices to low energy expense cost of the system. After the proposed matching game algorithm for TE allocation and minimum cost algorithm based on energy decision, the energy expense cost of the system was 115.31 RMB, which was approximately 24% lower than that of the terminal algorithm (86.52 RMB).

Subsequently, the energy expense costs of the system by using the proposed mixed energy supply algorithm under different numbers of TEs were observed.

Figure 5 shows the differences between the total energy consumption of the initial system under different numbers of TEs with the traditional algorithm of distance-based TE allocation and the proposed matching game algorithm for TE allocation. When the system used traditional algorithm, TEs were allocated simply according to their distances to SBSs, without considering the exchange of TEs among SBSs. When TEs were served by different SBSs, the desired transmitted powers were different, influencing the total energy consumption of the system. However, the proposed matching game algorithm for TE allocation can lower the transmitted power of the SBS by exchanging the serving SBS while assuring QoS of TEs. The proposed matching game algorithm considered the preferences of SBS to transmitted power of TEs through simulation results, thereby enabling the lower total energy consumption of the system while assuring QoS of TEs.

Figure 6 shows the energy expense costs of the system after using the traditional and mixed energy supply algorithms under different numbers of TEs. The proposed mixed energy supply algorithm first decreased the total energy consumption of the system by using the matching game algorithm for TE allocation. Later, this algorithm applied the minimum cost algorithm based on the energy decision when energy suppliers with different prices existed to change the energy purchasing ratios of SBSs from suppliers, aiming to decrease the total energy expense cost of the system. According to the simulation results, the proposed algorithm achieved a lower energy expense cost of system than the traditional algorithm. Taking K=400 as an example, the energy expense cost of the proposed algorithm was 169.77 RMB, which was approximately 52% lower compared with that of the traditional algorithm (359.41 RMB).

## 6. Conclusions

This study mainly discusses the energy expense cost optimization problem of the IoT system. A bilateral random model of TE allocation and energy decision in the IoT is designed by targeting the minimum energy expense cost of the IoT system, and a mixed energy supply algorithm based on the matching game and convex optimization is proposed. This algorithm is composed of two parts. First, the algorithm models the TE allocation in the IoT into a multiple-to-one matching game problem. Different preferences of SBSs and TEs, including gains from SBS service to TEs, transmitted power of SBSs to TEs, channel gain that TEs gained from SBSs, and service charges that TEs must pay for, are introduced. In this way, the TE-matching scheme with the minimum total energy consumption of the system is gained while meeting the QoS constraint of different TEs. Second, the proposed algorithm considers energy suppliers with different energy prices and maximum energy supply amounts in the smart grid and gains the energy purchase scheme for SBSs to minimize the energy expense costs of the system through the convex optimization algorithm. According to the simulation results, when K = 300, the proposed mixed energy supply algorithm in this paper reduces the total energy consumption of the system by about 42% and the energy cost of the system by 46%. In future, we will simulate and test the algorithm in the Software Defined Radio (SDR) based on the radio frequency integrated chip and Field Programmable Gate Array (FPGA).

## Figures and Tables

**Figure 1 sensors-20-05458-f001:**
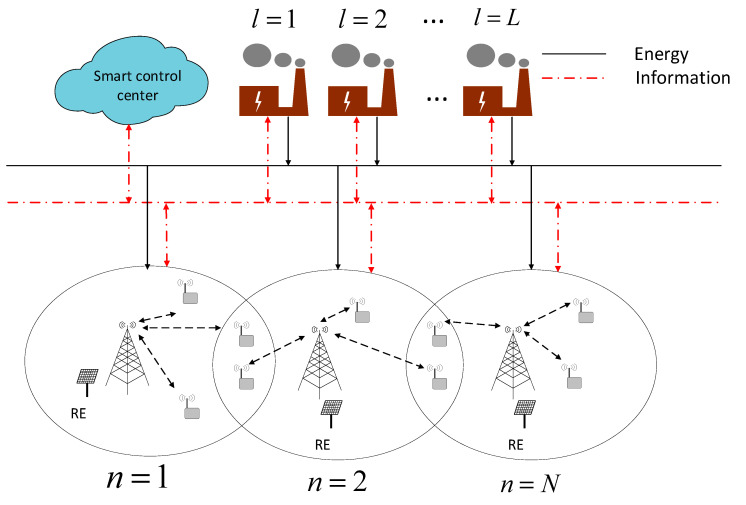
System model.

**Figure 2 sensors-20-05458-f002:**
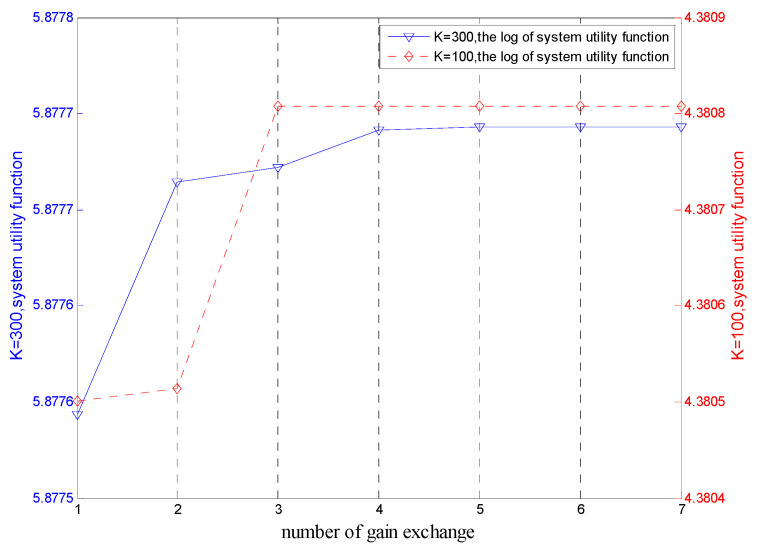
Variation curves of system utility functions at $K_{1}=100$ and $K_{2}=300$.

**Figure 3 sensors-20-05458-f003:**
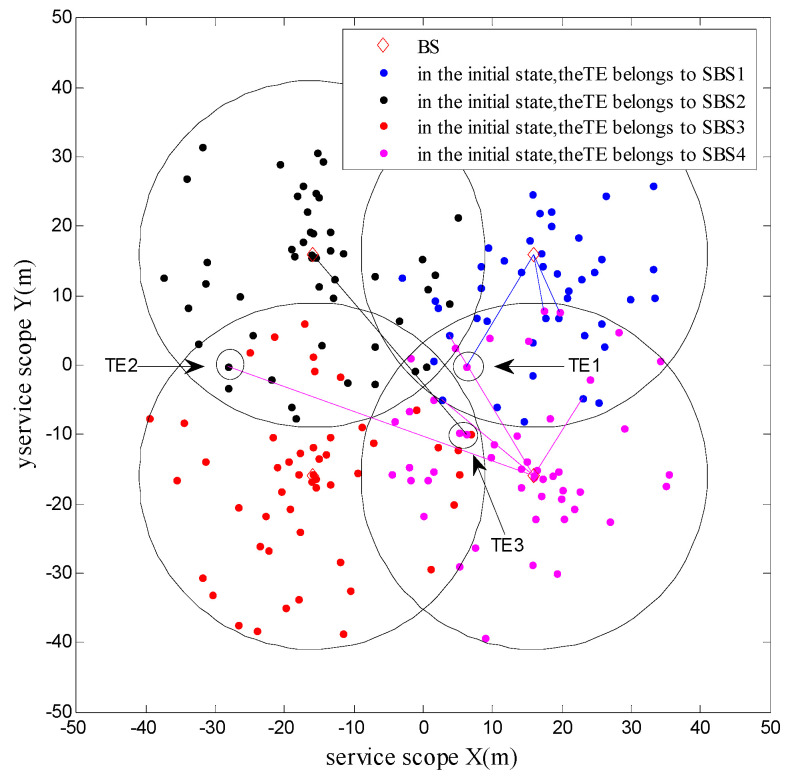
Terminal equipment allocation after minimum cost algorithm based on energy decision under K=200.

**Figure 4 sensors-20-05458-f004:**
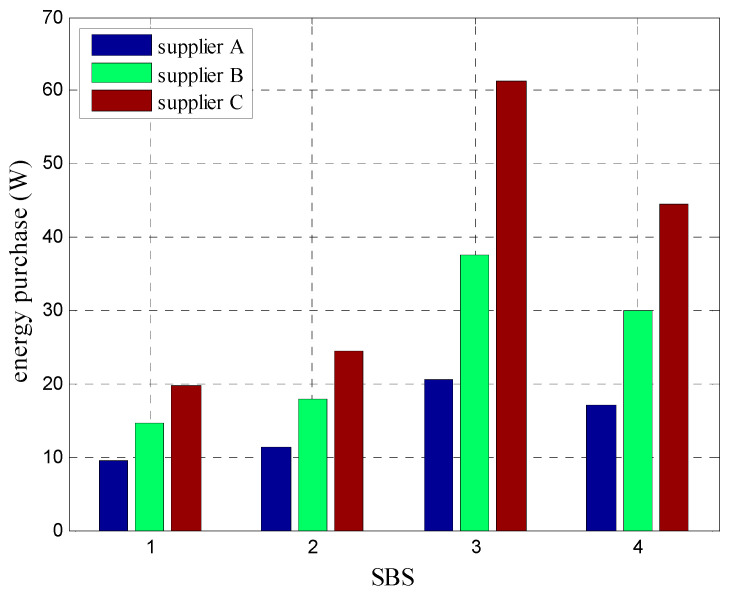
Energy purchase schemes of BSs under K=200.

**Figure 5 sensors-20-05458-f005:**
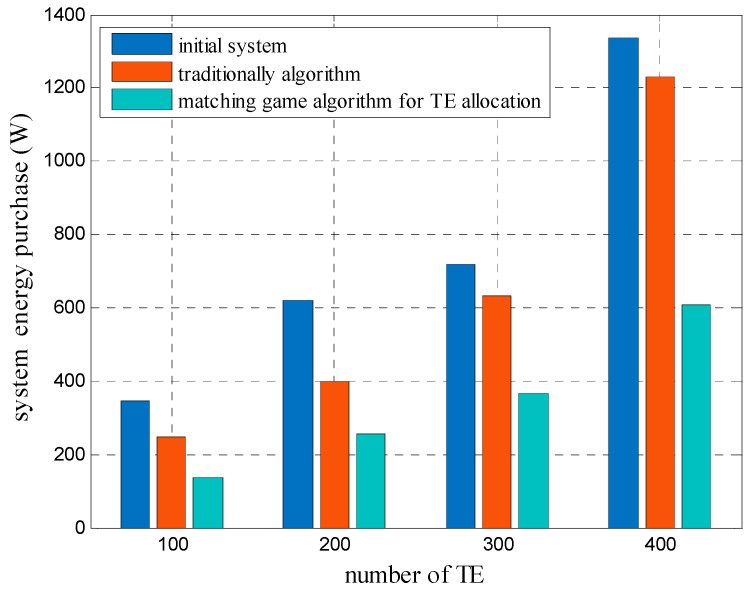
Energy consumption of the system under different numbers of terminal equipment.

**Figure 6 sensors-20-05458-f006:**
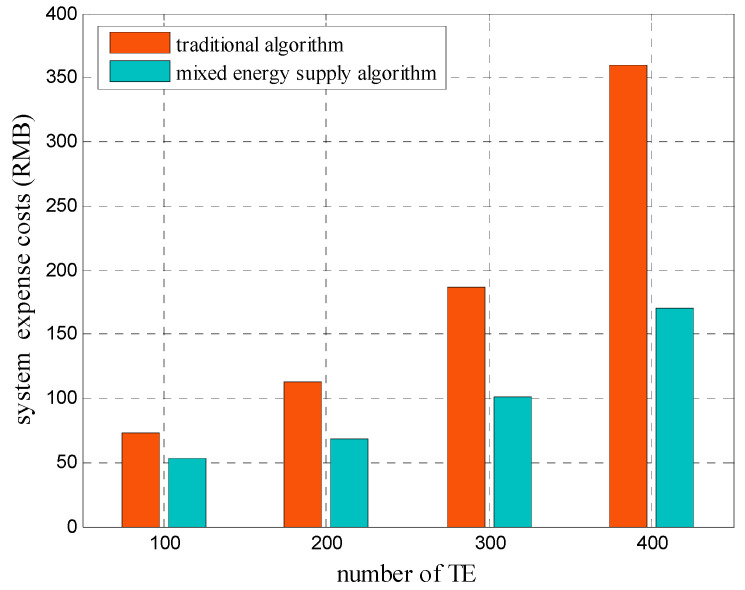
Comparison of energy expense costs of a system under different numbers of terminal equipment.

**Table 1 sensors-20-05458-t001:** Simulation parameters.

Parameters	Values
Service radius of SBSs	25 m [33]
Gains of transmitting and receiving antennas (G)	10 [34]
Gaussian noise power spectral density ($n_{0}$)	−50 dBm/Hz [34]
Channel bandwidth (B)	100 MHz [35]
Maximum power limit of SBSs ($P_{\max}$)	200 W [34]
Renewable energy generation capacity of SBSs ($g_{RE}$)	30 W
Bit rate of terminal equipment ($R_{k}$)	200–300 Mbps [36]
Charging price of SBSs ($\gamma_{n}$)	0.2–0.3 RMB
Energy suppliers (L)	3
SBSs (N)	4

**Table 2 sensors-20-05458-t002:** Number of gain exchange to reach stable matching under different numbers of TEs.

Number of TEs	Average Number of Gain Exchanges	Run Time (s)
100	3.4	0.816
200	4.3	3.081
300	5.5	8.781
400	7.8	19.586

**Table 3 sensors-20-05458-t003:** Parameters of energy suppliers.

Energy Suppliers	Maximum Energy Supply Amounts	Energy Price
A	75 W	0.34 RMB/W
B	100 W	0.21 RMB/W
C	150 W	0.30 RMB/W

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
