# Peer review of "Optimization of Mixed Energy Supply of IoT Network Based on Matching Game and Convex Optimization"

_sensors, 2020, doi:10.3390/s20195458_

Round 1
Reviewer 1 Report
I find the paper to be of interest in the current economical and technical context.
I also appreciated the approach, that I think has the potential to bring real improvements in terms of base station power efficiency, as well as autonomy in emergency situations.
However, while the proposed model is sound, the simulation presented in chapter 5. Results are seriously flawed:
- line 387 - all the BS have the same position (possible a writing error)
- service radius 25m - not realist for a real-world scenario
- "gains of .." - not clear from a technical point of view
- Bit rate 1-4 bit - not relevant for 5G applications
Considering the above, I find the test to be irrelevant for simulating IoT based communication in 5G scenarios.
I would like to see a simulation with relevant parameters.
Reviewer 2 Report
In this paper, the authors discuss the energy expense cost optimization problem of the IoT system, proposing two approaches as potential solutions: a bilateral random model of TE allocation and energy decision to target the minimum energy expense cost of the IoT system; and a mixed energy supply algorithm based on the matching game and convex optimization.
Although the manuscript is sometimes hard to follow, it has merit and thorough explanation and simulation. I would like to suggest some minor comments:
- The manuscript needs proofreading, there are several typos and grammar mistakes, specially in the Introduction section. Just some examples: *will bringing (line 31), Communication (line 37), .massive (line 40), *[1515] (ine 67), etc.
- I think that in the first paragraph of introduction needs a lot of reference to support all the statements and examples included in this paragraph.
- In line 407, authors say that the number of iterations to reach convergence is "acceptable". Acceptable based on what? Please explain or give examples or refs to support that this is "acceptable".
- In the Conclusions, i recommend including some quantitative values derived from results in terms of lowering the energy consumption compared to traditional systems (for instance, include in what % is lowered).
Reviewer 3 Report
As the energy consumption of the base stations is a critical communication cost factor for the 5G technology, the authors are presenting a combined method of optimizing the energy needs of these base stations. The proposed method takes into account both the QoS requirements of the terminals as well as the diverse billing policies of the energy suppliers available for covering the needs of the base stations. This new work being presented is well-documented and based on authors’ previous research findings. Nevertheless, some reworking is required to give it the perfect form for publication.
Changes are required, related with typos and English style corrections. In some cases a rephrasing is needed (e.g., at lines 17-18). In general, the authors should slightly further elaborate their style in order to make the description of their objectives and methods more apparent and clear.
In introduction: Agreed that the 5G concept has the lion’s share from the IoT pie, but they should also mention other important technologies used for IoT solutions like LoRa, ZigBee, Wi-Fi, etc. and justify their focus in 5G systems.
The paper becomes very technical quite early, i.e., by the beginning of section 2, and thus, without sacrificing the quality of their scientific description, the authors should enrich their article with more linking material among the different parts of their documentation. Further description of the interaction among the software modules being developed to implement the discussed algorithms, references to these tools and the mathematical methods and indicative code parts should be also welcome.
The authors provide a satisfactory set of measurements verifying the importance of their approach. In line 406 they say “Moreover, the number of iterations to reach convergence is acceptable”, it would help to stress out that the vertical axis, in blue, corresponds to the number of the iterations of the proposed algorithm. They should also clarify the amount of time and the machine type used to provide these satisfactory converging results. Finally, can the authors estimate if the whole algorithmic approach can be ported on an embedded system (of similar potential to the raspberry pi) hosted by the base stations?
Round 2
Reviewer 1 Report
I appreciate the modifications, but the 25m radius for the base station coverage is still not realistic for 5G. The cited article [29] does not refer to a typical 5G setup.
I think the article can be of interest but must either include a simulation that reflects 5G parameters, either clarify what it is simulating for (maybe just a WiFi network?).
Reviewer 3 Report
The authors provided an improved version that better highlights their work and efficiently incorporated most of the proposed improvements. They also made corrections in English language/style. Some issues are still present though, rising only minor suggestions.
a) In Introduction, the part being added that refers to other networking technologies acting complementary to the 5G one should be reworked and enriched in order to be less emphatic over the 5G case.
b) As the authors present a pure simulation study, it might be beneficial for the reader to mention that, in real-world conditions, the incurred handover operations of the mobile users between different networks or base stations comprise a costly and challenging process but, thankfully, many approaches the last decade are focusing in making these inevitable actions to happen more seamlessly. The following material could be beneficial towards this direction.
- Sen, Jaydip. (2010). Mobility and Handoff Management in Wireless Networks. 10.5772/8482.
- Dimitriou, N., Sarakis, L., Loukatos, D., Kormentzas, G. and Skianis, C. (2011). Vertical handover (VHO) framework for future collaborative wireless networks. Int. Journal of Network Management. 21. 548-564. 10.1002/nem.786.
- Seo DY, Chung YW. Modeling and Performance Evaluation of a Context Information-Based Optimized Handover Scheme in 5G Networks. Entropy. 2017; 19(7):329
- Lai WK, Shieh C-S, Chou F-S, Hsu C-Y, Shen M-H. Handover Management for D2D Communication in 5G Networks. Applied Sciences. 2020; 10(12):4409.
c) In Conclusions, to mention their intention for specific hardware testing and simulations in their future plans.
